# The relative impacts of tropical Pacific teleconnections and local insolation on mid-Holocene precipitation over tropical South America

Minn Lin Wong[1,2], Xianfeng Wang[1,2,3]

[1] Asian School of the Environment, Nanyang Technological University, Singapore
[2] Earth Observatory of Singapore, Nanyang Technological University, Singapore
[3] Center for Climate Change and Environmental Health, Nanyang Technological University, Singapore

*Correspondence to*: Minn Lin Wong (minnlin001@e.ntu.edu.sg)

**Abstract.** The study of El Niño–Southern Oscillation (ENSO)'s role in modifying past climates can provide valuable insights into the sensitivity of the present-day hydroclimate to changing ENSO characteristics. In this study, a water isotope-enabled atmospheric general circulation model (AGCM; ECHAM4.6) is utilized to examine the impact of two aspects of mid-Holocene ENSO characteristics on South American precipitation and precipitation $\delta^{18}O$: 1) Reduced amplitude of ENSO variability (relative to present-day) and 2) a mid-Holocene La Niña-like mean state in tropical Pacific sea-surface temperatures (SST), but with modern-day ENSO variability. Additionally, we conducted a lower Southern Hemisphere summertime insolation (SHSI) experiment to investigate the role of orbital forcing. Our results show that decreased ENSO variability results in a minor change in the climatological precipitation. In contrast, a La Niña-like mean state results in an east–west dipole pattern of precipitation change comparable to the effect of lower SHSI; however, the dipole pattern is not mirrored in precipitation $\delta^{18}O$. The experiments suggest that mid-Holocene western Amazon $\delta^{18}O$ is influenced by reduced SHSI via decreasing precipitation seasonality which drives positive $\delta^{18}O$ anomalies, while a La Niña-like mean state change suppress winter precipitation, leading to negative $\delta^{18}O$ anomalies. These opposing effects further highlight the additional role of global SST feedbacks in shaping western Amazon $\delta^{18}O$. In contrast, in the northeastern Amazon, both SHSI and a more La Niña-like ENSO mean state directly influence precipitation $\delta^{18}O$, resulting in strong negative $\delta^{18}O$ anomalies. Overall, these findings emphasize the combined effects of ENSO variability, tropical Pacific SSTs, and orbital forcing on South American precipitation and $\delta^{18}O$ during the mid-Holocene.

## 1 Introduction

### 1.1 Introduction and Background

The El Niño–Southern Oscillation (ENSO) is a mode of interannual climate variability within the coupled ocean-atmosphere system that has a significant impact on the global climate (Trenberth and Stepaniak, 2001). Through both a direct local forcing and far-field teleconnections, ENSO can have substantial effects across the South American continent by causing precipitation to deviate from the climatology (Cai et al., 2020). Therefore, changes in ENSO characteristics, such as its mean state and

spatio-temporal variability, can have significant ramifications on the climate of South America. As there is still a high degree of uncertainty in future projections of ENSO, the study of ENSO effects in the past using model simulations can provide

insights into the sensitivity of the South American climate to changes in different ENSO characteristics.

The mid-Holocene, defined by the Paleoclimate Modelling Intercomparison Project Phase Four (PMIP4) as 6 thousand years ago before present (kyr BP)) (Otto-Bliesner et al., 2017), serves as an intriguing timeframe for investigating how a change in ENSO characteristics can impact South American precipitation. During the mid-Holocene, a number of paleoclimate archives have documented that the magnitude of ENSO variability was lower than the modern-day (e.g., Chen et al., 2016; Emile-Geay

et al., 2016; White et al., 2017), and its temporal frequency was also reduced (Chen et al., 2016). Concurrently, the sea-surface temperature (SST) gradient across the tropical Pacific was stronger, producing a more 'La Niña-like' mean state (e.g., Barr et al., 2019; Brown et al., 2020). In addition to the changes in ENSO characteristics, the Southern Hemisphere summer insolation (SHSI) was significantly lower during the mid-Holocene (about 30 W/m$^2$ difference at 10°S (Laskar et al., 2004)) and thus can also result in precipitation changes over South America due to a weaker land-ocean thermal gradient during the austral summer

(December to February, DJF). Mid-Holocene precipitation over South America is therefore affected by the compounded effects of external forces (directly by insolation change) and internal variability (via changes in ENSO and its atmospheric teleconnections to the climate system of South America).

In this study, we make use of a water isotope-enabled Atmospheric General Circulation Model (AGCM), the ECHAM4.6, to characterize how a change in ENSO characteristics (variability and mean state) and insolation could have affected the

precipitation response in South America during the mid-Holocene. In the real climate system, ENSO mean state, frequency and variability are dynamically linked and can covary (e.g., Chung and Li, 2013; Lübbecke and McPhaden, 2014). The use of single-forcing experiments within an AGCM, however, allows us to separate these components and conduct controlled sensitivity tests to understand their first-order influence on South American precipitation and isotopic responses. While this approach necessarily simplifies the coupled feedbacks of the ocean-atmosphere system, it provides a clear approach to identify

the principal pathways and relative contributions through which ENSO-related forcings may shape the regional hydroclimate. The effects of a mid-Holocene ENSO state are addressed with two key questions: 1, How does a change in ENSO variability shift the mean climatology of precipitation over tropical South America? And 2, How does a mean state change, but with the same ENSO variability, shift the mean climatology of precipitation over the region? Furthermore, we compare the simulated oxygen isotopic composition ($\delta^{18}O$) with regional $\delta^{18}O$ proxy records. This comparison aids in illuminating the primary factors

influencing $\delta^{18}O$ records and provides a comprehensive understanding of the mechanisms driving hydroclimate changes during the mid-Holocene.

## 1.2 Mid-Holocene ENSO variability and mean state

Since ENSO is spatially and temporally variable, it is important to first define different ENSO characteristics. In this study, we focus on two key characteristics of ENSO — the ENSO variability (amplitude and frequency) and the mean state. The

amplitude of ENSO variance is defined as the departure of SSTs from the climatological mean, expressed by the standard

deviation of SSTs (Fig. 1), while the temporal variability has been defined as the periodicity of ENSO events (Moy et al., 2002; Rodbell et al., 1999).

The 'mean state' refers to the climatological SST of the tropical Pacific, which encompasses the seasonality and spatial pattern of SSTs including the zonal SST gradient. As the mean state refers to the SST over a period of time, changes in the mean state can be caused by changes in insolation or atmospheric greenhouse gas concentrations. By definition, ENSO is evidenced by SSTs fluctuating around a zero mean; therefore, a change in the ENSO amplitude can theoretically occur without a change in the mean state (e.g. blue curve in Fig. 1), and a change in the mean state can occur without changing the variability (e.g. green curve in Fig. 1). Although this is rarely the case in the real world, an AGCM would allow us to separately test the impacts of either characteristic on the South American climate.

Numerous studies have shown that the mid-Holocene registered a change in both the ENSO variability (which was suppressed) and in the mean state (which was more 'La Niña–like'). Marine and terrestrial proxy records (e.g., corals, molluscs, marine sediments, and speleothems) provide evidence of a reduction of ENSO variability of 60% (Chen et al., 2016; Cobb et al., 2013; Emile-Geay et al., 2016; Grothe et al., 2020; Koutavas and Joanides, 2012). Fully coupled climate models also corroborate the reduced variability (Brown et al., 2020; Pausata et al., 2017), albeit with a typical reduction of ~20%, which is less than that recorded in the proxies. Likewise, terrestrial archives in South America and Australia (Barr et al., 2019; Carré et al., 2012) and climate model simulations (Brown et al., 2020; Koutavas and Joanides, 2012; Shin et al., 2006) demonstrate an increased SST gradient across the tropical Pacific, supporting a more 'La Niña–like' mean state of the tropical Pacific compared to the present-day.

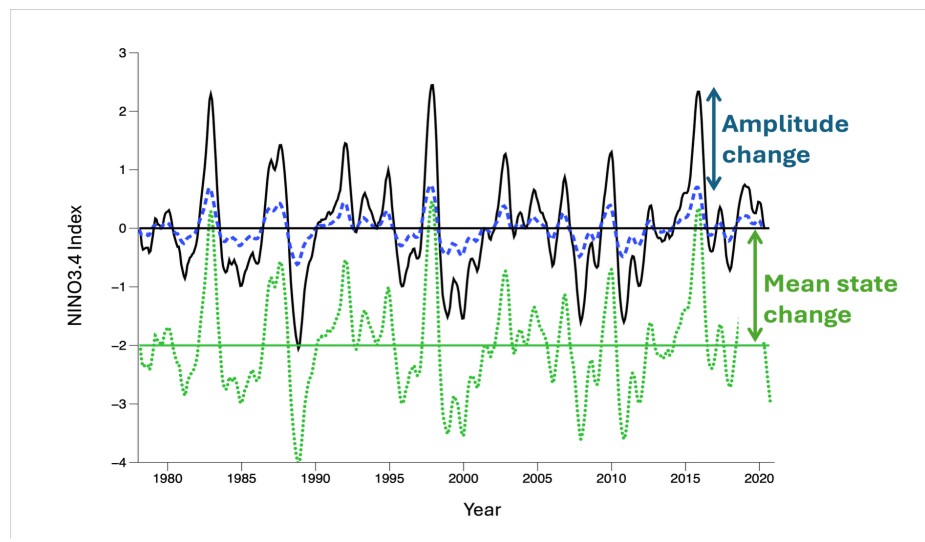

**Figure 1: Illustration of changes in the mean state and ENSO characteristics. In this study we consider two mid-Holocene scenarios based on the paleo records: a change in the amplitude of ENSO anomalies (blue dashed curve) with no change in the mean state (NoENSO, black horizontal line), and a shift in the mean state (LNstate, green horizontal line) without any change in ENSO amplitude (green dotted curve). ENSO is tracked by the Nino3.4 Index (black curve) relative to the climatological mean (black**

 horizontal line). Note that ENSO anomalies are defined by departures from the climatology. Hence, the mean states (horizontal lines) in these two experiments include a climatological seasonal cycle.

## 1.3 South American hydroclimate during the mid-Holocene

Paleo-hydroclimate records in tropical South America suggest a broad northeast–southwest dipole pattern of anti-phased precipitation trends over the mid to late-Holocene. Figure 2 shows a compilation of the $\delta^{18}O$ records from speleothems, ice cores and authigenic lake carbonate archives that use $\delta^{18}O$ as a proxy for precipitation change (details of records in Sect. S2 in the Supplement). Records from the western Amazon, Andes, and Southern Brazil mostly indicate a drier mid-Holocene climate than the present day, while the records in the eastern Amazon and Nordeste region show a wetter mid-Holocene climate. Although the northeast–southwest dipole pattern is substantiated by other types of climate proxies, such as pollen from sediment records (Gorenstein et al., 2022), here we chose to focus on $\delta^{18}O$ proxies as these can be directly compared with the $\delta^{18}O$ output from the model simulation.

The east–west anti-phased response in precipitation is conventionally explained by changes in the local zonal circulation system. During the austral summer, strong latent heating over the Amazon typically generates a Rossby wave response to the west, establishing the 'Bolivian High' as an upper-tropospheric anti-cyclone over northern Argentina, and a Kelvin wave response eastwards, known as the 'Nordeste Low', that results in subsidence over the tropical Atlantic and creates arid conditions over the northeastern states of Brazil (Lenters and Cook, 1997; Silva Dias et al., 1983). During the mid-Holocene, the lower SHSI led to a weaker South American Summer Monsoon (SASM) convection over the western Amazon. This generated a corresponding reduction in the strength of the Nordeste Low over the tropical Atlantic, consequently reducing the aridity over northeastern Brazil. As a result, drier conditions prevailed over the west and wetter conditions occurred over the east during the mid-Holocene (Cheng et al., 2013; Cruz et al., 2009). However, this explanation does not consider the influence of changes in ENSO characteristics or the tropical Pacific climatology during the mid-Holocene, which has also been noted to contribute to contrasting precipitation responses across regions in the present-day climate (Cai et al., 2020).

In this study, we investigate the role of changing ENSO characteristics and insolation on tropical South American precipitation during the mid-Holocene. The methodology for conducting the AGCM experiments and the approach for calculating $\delta^{18}O$ values are outlined in section 2. In section 3, we evaluate the performance of the AGCM in simulating modern precipitation, precipitation $\delta^{18}O$ and ENSO. We additionally describe the results from each of the model experiments compared to a modern-day control run. Section 4 discusses the impact of ENSO variability, mean state, and SHSI on mid-Holocene precipitation and $\delta^{18}O$, drawing insights from the AGCM simulation results and comparisons with regional paleorecords. Finally in section 5, we summarize the key finding of this study and significance of this research.

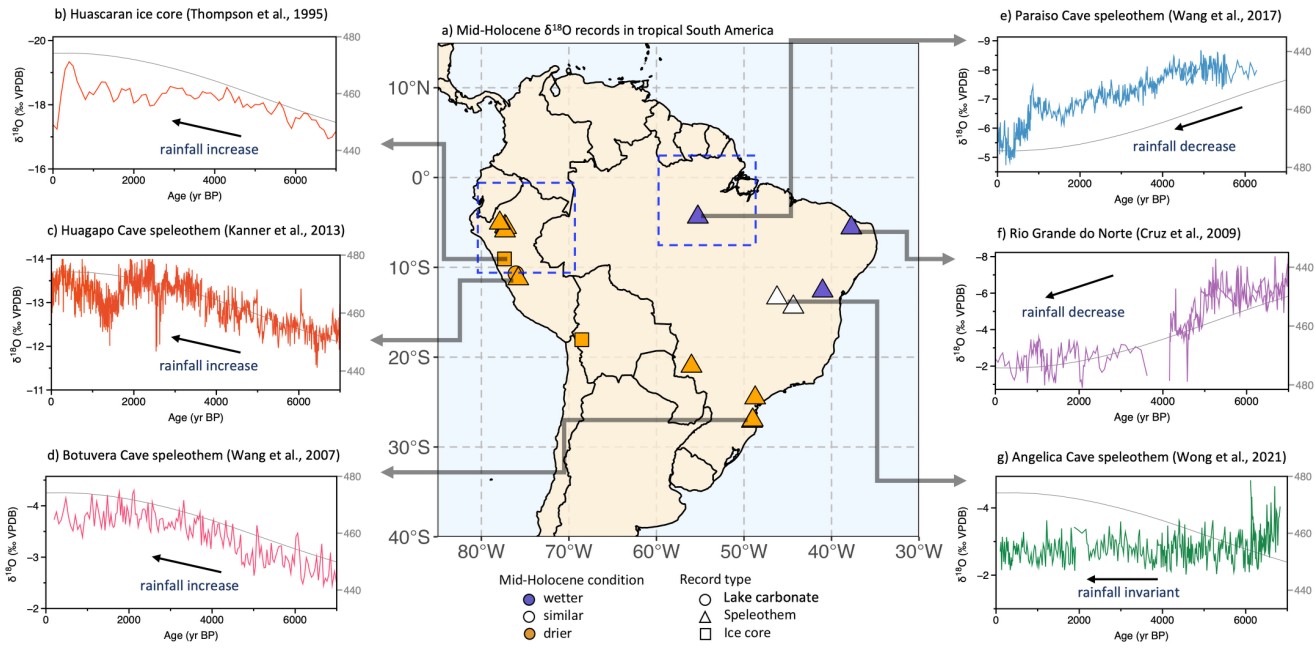

**Figure 2: Paleo-hydroclimate $\delta^{18}O$ records from South America and their relationship with local summer insolation over the mid-to-late Holocene. a) Location of individual $\delta^{18}O$ records that span the mid-Holocene to pre-industrial from speleothems (triangle), or ice cores (squares), and authigenic lake carbonates (circles). Drier, similar-to-present and wetter mid-Holocene conditions are indicated by the orange, white and purple colours, respectively. $\delta^{18}O$ records are shown for: b) Huascarán Ice core (Thompson et al., 1995); c) Huagapo Cave (Kanner et al., 2013); d) Botuvera Cave (Bernal et al., 2016; Wang et al., 2007); e) Paraiso Cave (Wang et al., 2017); f) Rio Grande do Norte Caves (Cruz et al., 2009), and g) Angelica Cave (Wong et al., 2021). $\delta^{18}O$ is plotted in a reversed scale. For panels b–g, the January insolation values at 15°S are shown by the grey lines (Laskar et al., 2004). Note that the insolation curve is plotted in reverse scale for panels e–f to facilitate comparison with $\delta^{18}O$ trends. Locations of individual records as listed in Table S2. The dashed boxes in panel (a) show the region used for the seasonality analysis in Fig. 6 of western Amazon and eastern Amazon.**

## 2 Methodology

### 2.1 The model

The AGCM used in this study is the ECHAM model version 4.6 (ECHAM4.6 (Roeckner et al., 1996)). The ECHAM4.6 is based on a hybrid sigma-pressure coordinate system and was run with triangular truncation at a wave number of 42 (T42, 2.78° lat. × 2.78° long.) including 19 vertical layers of atmosphere. The model is additionally equipped with a water isotope module to simulate the distribution of water isotopes throughout the atmosphere (Hoffmann et al., 1998). The ECHAM is a popular model utilized in exploring the atmospheric response to various forcings (e.g., Battisti et al., 2014; Jiang et al., 2020; Samanta et al., 2019) and has demonstrated the capability of capturing mid-Holocene $\delta^{18}O$ patterns (Werner, 2019). Here we run the model with prescribed SSTs, discarding the first year to avoid any equilibration problems during the model spin-up.

## 2.2 Experimental Design

We run five AGCM experiments to isolate the effects of different climate forcings, summarized in Table 1. While the use of an AGCM necessarily omits the full dynamical coupling between ocean and atmosphere, it provides a straightforward means to isolate key mechanisms and evaluate the direction and relative strength of atmospheric responses. The experiments should therefore be viewed as targeted sensitivity analyses that show first-order impacts rather than as fully coupled equilibrium simulations.

The 'Control' experiment is forced by the observed 41-year history of (1979–2019) monthly averaged SST derived from the ERA5 Reanalysis dataset (Hersbach et al., 2020) (interpolated to the model time step), and with prescribed modern-day boundary conditions, greenhouse gases and orbital parameters. As such, the Control experiment includes the impact of the full modern day ENSO variability.

The second experiment ('MidH') represents a mid-Holocene scenario. In this experiment, the model is forced by mid-Holocene insolation and greenhouse gas concentrations. Within the tropical Pacific (160–275°E, 20°N–20°S), we prescribe a climatological SST based on the ensemble mean of mid-Holocene simulations from PMIP4 (for details, see Sect. S1 in Supplement), with a linear taper of 5.6° in both latitude and longitude to ensure a smooth transition to surrounding SSTs. Hence, the MidH simulation has no ENSO variability. As we will show, changes in the character of ENSO have little impact on the climatological precipitation or $\delta^{18}O$ of precipitation over tropical South America (see 'NoENSO' experiment below). Therefore, we can directly compare the change in the precipitation weighted $\delta^{18}O$ ($\delta^{18}O_p$) over tropical South America in the 'MidH' and 'Control' experiments to the change in $\delta^{18}O$ recorded in the speleothems.

To isolate the residual impact of ENSO on the climatological annual cycle of precipitation and $\delta^{18}O_p$ over tropical South America, we perform the 'NoENSO' experiment. The NoENSO experiment is identical to the Control experiment, only the SST over the tropical Pacific is replaced by the climatological SST (1979–2019). The difference between the NoENSO and Control experiments shows the rectified impact of ENSO on the climatology of precipitation and $\delta^{18}O$ of precipitation.

As in the MidH experiment, the 'LNstate' experiment uses mid-Holocene values of orbital parameters and greenhouse gas concentrations and is forced by the mid-Holocene SST taken from the PMIP4 ensemble superimposed on modern day ENSO anomalies (1979–2019) in the tropical Pacific. Since proxy data and modelling studies of the mid-Holocene climatological SST features a La Nina-like cooling in the eastern and central tropical Pacific compared to the modern day, this experiment is called LNstate. Comparing the outputs from the LNstate to the Control simulations isolates the impact of the changed climatological annual cycle with no change in ENSO characteristics.

In the 'MHinsol' experiment, the model is forced by the same SST as in the Control experiment, but with mid-Holocene orbital values and greenhouse gas concentrations. Changes in insolation impact the climate over tropical South America directly by modifying land–ocean thermal gradients, and indirectly through insolation-induced changes in the tropical Pacific SST and the subsequent atmospheric teleconnections to tropical South America (Cai et al., 2020). Comparing the results

from the Control experiment to the MHinsol experiment illuminates the impact of local (direct) insolation changes on South
American precipitation and the $\delta^{18}O$ of precipitation. Table 1 summarizes the five experiments.

| | | 'Control' | 'MidH' | 'NoENSO' | 'LNState' | 'MHinsol' |
|---|---|---|---|---|---|---|
| Orbital | Eccentricity | 0.016724 | 0.018682 | 0.016724 | 0.016724 | 0.018682 |
| | Obliquity | 23.446 | 24.105 | 23.446 | 23.446 | 24.105 |
| | Perihelion | 100.33 | 0.87 | 100.33 | 100.33 | 0.87 |
| GHG | $CH_4$(ppbv) | 808.3 | 597.0 | 808.3 | 808.3 | 597.0 |
| | $CO_2$(ppmv) | 284.3 | 264.4 | 284.3 | 284.3 | 264.4 |
| | $N_2O$(ppbv) | 273.0 | 262.0 | 273.0 | 273.0 | 262.0 |
| | CFC-11 & 12 (ppbv) | 0 | 0 | 0 | 0 | 0 |
| Tropical Pacific SST | Mean state | Modern | Mid-Holocene climatology | Modern | Mid-Holocene climatology | Modern |
| | Variability | Modern | No ENSO variability | No ENSO variability | Modern | Modern |

Table 1: Experiment scenarios using the ECHAM4.6 water isotope enabled model. The model was run in a prescribed SST mode
forced with 41-years of SST data (1979–2019; based on the ERA5 Reanalysis dataset). For ENSO experiments, the modified SSTs
are imposed over the central to eastern tropical Pacific region (160–275°E; 20°N–20°S). Solar activity, topography, ice-sheets and
coastlines are the same between the two scenarios.

## 2.3 Method for attributing $\delta^{18}O$

The precipitation-weighted $\delta^{18}O_p$ from the model is compared to the $\delta^{18}O$ in the speleothems and to the observed $\delta^{18}O_p$ from
the Global Network of Isotopes in Precipitation (GNIP) (IAEA/WMO, 2021). The precipitation-weighted $\delta^{18}O$ is shown in
Eq. (1):

$$\delta^{18}O_p = \frac{\sum_m \delta^{18}O_m \cdot P_m}{\sum P_m}, \tag{1}$$

where the $\delta^{18}O_m$ is the monthly isotopic composition of precipitation, Pm is the monthly precipitation amount and m is the
number of months, m = 480 ( = 12 months/year × 40 years).

Thus, changes in the $\delta^{18}O_p$ can be decomposed into two parts: 1) the change in the isotopic composition of the precipitation
itself which can be a result of precipitation intensity (i.e., the local 'amount effect') or other changes in the ambient vapour

$\delta^{18}O$ that condenses (e.g., by moisture source change); and 2) the change in the monthly amount of precipitation which has to do with the change in precipitation seasonality (Liu and Battisti, 2015).

The change in $\delta^{18}O_p$ caused by a change in the annual cycle of $\delta^{18}O$ of precipitation is shown in Eq. (2):

$$\frac{\sum_m \delta^{18}O_{m,E} \cdot P_{m,E}}{\sum P_{m,E}} - \frac{\sum_m \delta^{18}O_{m,C} \cdot P_{m,E}}{\sum P_{m,E}}, \qquad (2)$$

where subscript 'C' refers to the 'Control' experiment and subscript 'E' refers to one of the other four experiments. The change in $\delta^{18}O_p$ caused by a change in the monthly precipitation amount is shown in Eq. (3):

$$\frac{\sum_m \delta^{18}O_{m,E} \cdot P_{m,E}}{\sum P_{m,E}} - \frac{\sum_m \delta^{18}O_{m,E} \cdot P_{m,C}}{\sum P_{m,C}}, \qquad (3)$$

**3 Results**

**3.1 Performance of ECHAM4.6 in simulating modern-day precipitation and precipitation $\delta^{18}O$**

The average precipitation pattern during each season for the 'Control' simulation is compared with the ERA5 reanalysis data over the same time-period (1980–2019) (Fig. S3). The model captures the essential features of South American climate, such as the strong SASM convection over the Amazon and the manifestation of the South Atlantic Convergence Zone (SACZ)

during the austral summer months. The model also simulates the annual latitudinal migration of the Intertropical Convergence Zone (ITCZ).

The precipitation weighted $\delta^{18}O$ ($\delta^{18}O_p$) from the Control experiment captures the zonal gradient of $\delta^{18}O_p$ seen in the GNIP stations (Fig. S4a) which stems from the 'continental effect' (Vuille and Werner, 2005) — the least negative values occur along the northeastern coast while the most negative values occur inland, in western Amazon and along the Andes. However,

the simulated gradient in $\delta^{18}O_p$ from the model is weaker than that seen in GNIP stations, likely a result of too weak convection over the western Amazon and Andes related to a coarse resolution of the model. A comparison with a higher resolution setup (T106) shows a steeper continental gradient in $\delta^{18}O_p$ (Fig. S4b), suggesting that the coarser resolution likely underestimates the $^{18}O$ depletion towards the west due to the inability to capture the full orographic effect towards the Andes. While this resolution-dependant bias is inherent to the model setup, our approach is to focus on the anomalies between the

experiments and the baseline. This will allow us to highlight scenario-driven changes, rather than absolute values, and thus, the $\delta^{18}O_p$ differences can be interpreted robustly.

**3.2 Performance of ECHAM4.6 simulating ENSO**

To evaluate how realistically the model captures the ENSO impacts on precipitation over South America, the correlation

between the Niño 3.4 index and monthly precipitation anomalies over South America is compared between the ECHAM4.6 simulation and ERA5 Reanalysis data (Fig. S5a–b). Both datasets show a significant positive correlation with Niño 3.4 in southeastern South America, where a warm (cold) Niño 3.4 phase corresponds to positive (negative) precipitation anomalies. A negative correlation between Niño 3.4 and precipitation anomalies is also significant in northeastern Brazil and eastern

Amazon. However, the western Amazon shows no significant correlation between precipitation anomalies and Niño 3.4 at the 95% confidence level, suggesting other overlapping factors may be influencing ENSO-related signals.

The relationship between Niño 3.4 and $\delta^{18}O$ measured at GNIP stations as well as the model suggests that $\delta^{18}O$ over most of tropical South America becomes more positive (negative) during the warm (cold) Niño 3.4 phases (Fig. S5c–d), agreeing with previous modelling studies (Vuille et al., 2003). The association of monthly $\delta^{18}O$ with Niño 3.4 is logical since precipitation $\delta^{18}O$ is strongly correlated to precipitation amount in the tropics (Risi et al., 2008; Vuille et al., 2003). Therefore, ENSO should also exert some influence on $\delta^{18}O$ of precipitation. In the model results, the western Amazon shows a positive correlation between $\delta^{18}O$ and Niño 3.4, although there is no significant correlation with precipitation over certain areas likely due to an inherited $\delta^{18}O$ signature from upstream (northeast and eastern Amazon).

To verify that the ECHAM4.6 captures the effects of each ENSO phase on the South American climate, we follow the method by Vuille et al. (2003) to observe the austral summer (December to February, DJF) and winter (July to August, JJA) precipitation difference between composites of El Niño and La Niña years (Fig. S5e–h). In DJF, the reduction in precipitation over northern parts of South America (north of the equator) caused by El Niño is well reproduced by the ECHAM4.6. In JJA, El Niño causes a surplus of precipitation over most of the mid-latitudes of South America, consistent with evidence from Argentina (e.g. Montecinos et al., 2000) and southern Brazil (e.g. Grimm et al., 2000), but a reduction in precipitation mostly over the northeastern region, although the latter is slightly overestimated in the ECHAM4.6 simulation.

## 3.3 Precipitation results from the ECHAM4.6 simulations

The difference between the 'MidH' and 'Control' simulation, referred to as 'ΔMidH' should closely approximate the mid-Holocene conditions as captured by regional proxy records. Figure 3a shows that precipitation increases over the eastern Amazon and tropical Atlantic Ocean along the Nordeste states, while decreasing over the western Amazon and southern Brazil. This east–west dipole pattern agrees with the indicated direction of precipitation change suggested by hydroclimate records.

The difference between the 'LNstate' and 'Control' simulations, denoted as 'ΔLNstate' (Fig. 3b) and the difference between the 'MHinsol' and 'Control' simulations, denoted as 'ΔMHinsol' (Fig. 3c), both exhibit a broadly similar spatial pattern, with reduced precipitation extending from the western Amazon to southern Brazil. However, differences in precipitation response occur in the northeast, where ΔMHinsol precipitation increases primarily over the western tropical Atlantic Ocean and along the coastline of the Nordeste states, with little to no change over the eastern Amazon (Fig.3c). In contrast, ΔLNstate shows a significant increase in precipitation over the eastern Amazon (Fig. 3b).

Finally, the impact of ENSO variability is assessed through the difference between the 'NoENSO' and 'Control' simulations, denoted as ΔNoENSO (Fig. 3d). The absence of modern-day ENSO variability results in increased annual mean precipitation over northeastern South America and decreased precipitation over southern Brazil. However, these changes in both annual mean precipitation and precipitation $\delta^{18}O_p$ are relatively minor and do not reach the 95% confidence level across most of South America.

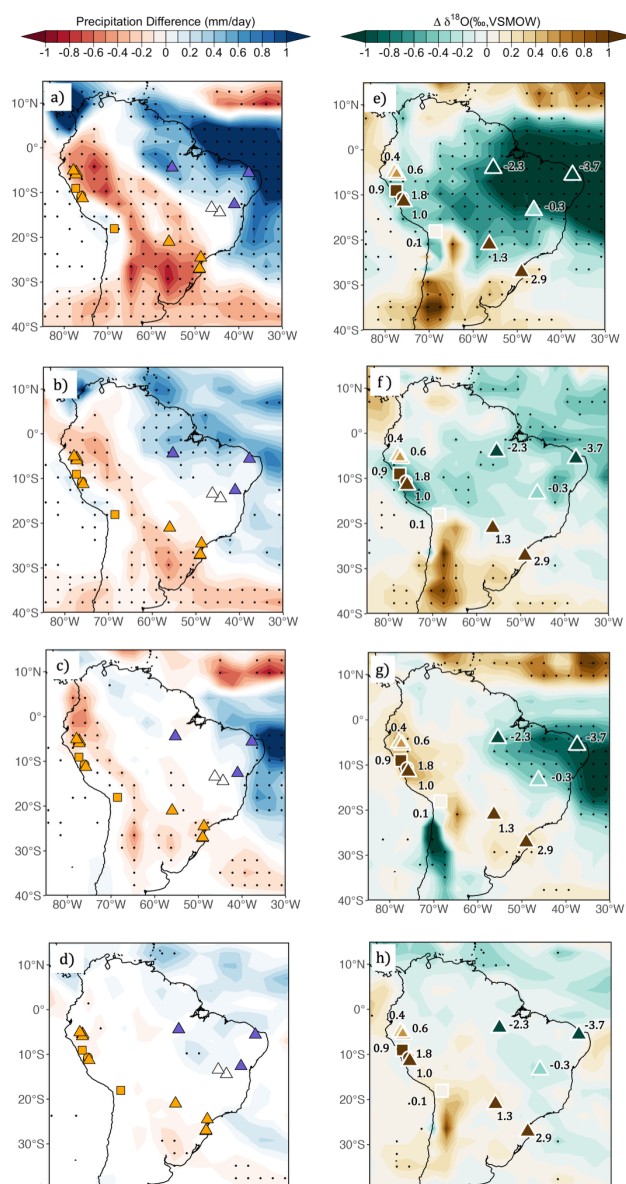

**Figure 3: The changes in precipitation and precipitation-weighted $\delta^{18}O$ ($\delta^{18}O_p$) relative to the Control experiment (experiment minus control). Shading shows the change in the mean annual precipitation rate in the (a) MidH, (b) LNstate, (c) MHinsol, and (d) NoENSO experiment, respectively. Second column: Same as the first column but for $\delta^{18}O_p$. The locations of hydroclimate proxy records are indicated by the coloured shapes, repeated from Fig. 2a. Numbers next to the proxy sites show the change of $\delta^{18}O_p$ value between the mid-Holocene and pre-industrial period (See Sect. S2 in the Supplement and Table S2 for calculating $\delta^{18}O$ of proxy records).**

## 3.4 Precipitation $\delta^{18}O_p$ results from the ECHAM4.6 simulations

Although the precipitation response in ΔMidH exhibits an anti-phased response between the northeast and western regions of the continent, the $\delta^{18}O_p$ output shows some discrepancy when compared to regional $\delta^{18}O$ proxy records (Fig. 3e). The ΔMidH response agrees well with records from the northeast, such as the Rio Grande do Norte record (Cruz et al., 2009), which show a large $\delta^{18}O$ decrease of up to 4‰. However, there is a clear mismatch between the ΔMidH simulation and $\delta^{18}O$ records in the western Amazon and Andes, where proxy data indicate positive $\delta^{18}O$ anomalies, in contrast to the muted modelled $\delta^{18}O$ response in the west. Therefore, while the 'MidH' scenario produces a northeast–southwest dipole pattern in precipitation, the dipole pattern is not reflected in the $\delta^{18}O_p$ values, suggesting that the $\delta^{18}O_p$ output contradicts the expected 'amount effect' over the western Amazon, where lower precipitation rates would typically be associated with more positive $\delta^{18}O_p$ values.

Examining the single-forcing experiments can provide insight into the causes of the overall $\delta^{18}O_p$ response. In ΔMHinsol the $\delta^{18}O_p$ output corresponds to the dipole in precipitation changes in that lower (higher) precipitation rates in the western Amazon and southern Brazil (northeast Brazil) produce more positive (negative) $\delta^{18}O_p$ values (Fig. 3g). In contrast, ΔLNstate shows broadly negative $\delta^{18}O_p$ anomalies across tropical South America, including the western Amazon (Fig. 3f). This suggests that the combined effects of both forcings contribute to a strong negative $\delta^{18}O_p$ response in the northeast while muting the $\delta^{18}O_p$ signal in the western Amazon. In contrast, ΔNoENSO produces only minor $\delta^{18}O_p$ response compared to ΔLNstate and ΔMHinsol. In the following sections, we will focus first on examining the relative impacts of mid-Holocene mean state and SHSI in section 4.1 to 4.3, while the impact of the modern-day ENSO on precipitation is discussed in section 4.4.

## 4 Discussion

### 4.1 Tropical Pacific mean state and insolation effects on zonal circulations

ΔLNstate and ΔMHinsol display a similar magnitude of precipitation change, resulting in the overall northeast–southwest dipole pattern seen in the ΔMidH. This suggests that the modifications in the mean state of the tropical Pacific may contribute equally to the spatial patterns of mid-Holocene precipitation change in South America, alongside the effects of insolation. To better illustrate the variations in spatial trends, the model results from these scenarios are compared to hydroclimate records in the region in Fig. 3b–c. The most notable difference in simulated precipitation occurs over the eastern Amazonian region, where the ΔLNstate shows higher mid-Holocene precipitation, agreeing with the Paraiso speleothem record located in eastern Amazon (Fig. 3b) (Wang et al., 2017). Conversely, ΔMHinsol exhibits no significant change compared to the control run (Fig. 3c).

To shed light on the different impacts of the SST mean state and insolation on the climate of South America, we examine the changes in zonal circulations that influence the region. In the ΔLNstate scenario, zonal circulation changes are observed in the Pacific Walker circulation (PWC). Figure 4a–b shows that there exists anomalous subsidence around the central Pacific at 170°E and enhanced rising motion over the South American continent at 60–40°W. This intensified ascending motion is

concentrated over the eastern Amazon region, leading to increased precipitation over a significant portion of northeastern South America during the months of DJF to March–April–May (MAM).

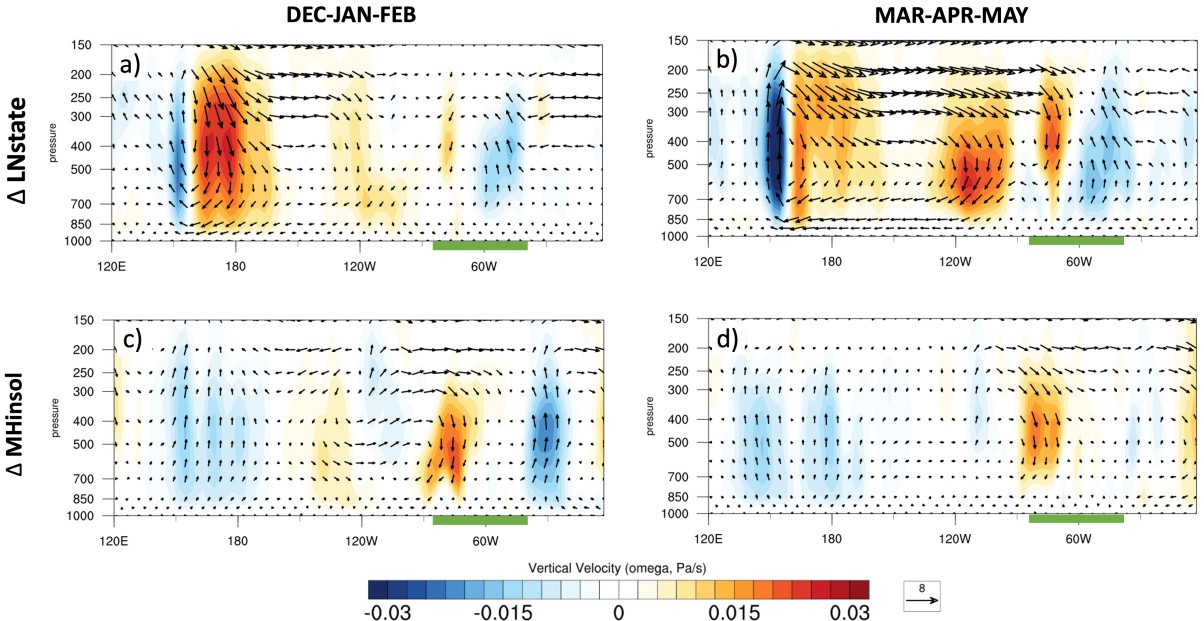

**Figure 4: Contours show the changes in the meridionally averaged vertical velocity (from 5°N–5°S) and vectors show u–w winds (m/s) for (a,b) ΔLNstate; and (c,d) ΔMHinsol. Positive vertical velocity values in the plot represent areas of anomalous subsidence and negative vertical velocity values as areas of anomalous rising motion. The region of South America is indicated by the green bar.**

On the other hand, zonal circulation changes in ΔMHinsol occur mainly over the South American continent and adjacent Atlantic Ocean in association with the local Bolivian High and Nordeste Low system. Enhanced subsidence over the South

American continent occurs strongest during the peak summer months in DJF (Fig. 4c). This is a result of the weaker Amazonian convection caused by reduced insolation over the continent that impairs the typical ascending motion over the continent at 70–80°W, thus leading to a weakening of the upper atmospheric Bolivian High. This, in turn, results in the ascending anomalies centred over 30°W due to the correspondingly weakened Nordeste Low. Therefore, unlike ΔLNstate, the anomalous rising motion in ΔMHinsol falls over the tropical Atlantic Ocean. This explains the spatial pattern of increased precipitation

concentrated over the western tropical Atlantic and northeastern Brazil that does not extend to the eastern Amazon, as shown in Fig. 3c. This is in contrast to ΔLNstate, which exhibits wetter conditions located further inland over northeastern South America and the eastern Amazon, as depicted in Fig. 3b.

Paleoclimate proxies have suggested that during the mid-Holocene, wetter conditions extended from the coast of northeastern Brazil (e.g., Cruz et al., 2009) to the eastern Amazon, up to 65°W (Prado et al., 2013; Wang et al., 2017). Our experiments

align with these findings, indicating that the conditions along the northeastern coast are primarily driven by insolation-induced

changes in zonal circulations over South America. However, to explain the precipitation change over the eastern Amazon, the changes in the mean state of the tropical Pacific and the subsequent alterations in the PWC are necessary.

## 4.2 The relative influence of a La Niña-like mean state and mid-Holocene insolation on precipitation $\delta^{18}O_p$

The $\Delta$MidH simulation shows that while the combined effects of a La Niña-like state and mid-Holocene insolation can result in the east–west dipole pattern of precipitation response, this pattern is not reflected in $\delta^{18}O_p$. Furthermore, although the dipole pattern in precipitation change is simulated by both $\Delta$LNstate and $\Delta$MHinsol, only $\Delta$MHinsol produces a corresponding dipole in $\delta^{18}O_p$ values that mirrors the regional $\delta^{18}O$ proxy records. The factors driving changes in overall $\delta^{18}O_p$ values can be analysed using Equations 2 and 3, which separate the contributions from variations in the annual cycle of precipitation $\delta^{18}O_p$ and changes in monthly precipitation amount, respectively.

Figure 5 illustrates the relative contributions of precipitation amount and precipitation $\delta^{18}O$ from the $\Delta$MidH, $\Delta$LNstate and $\Delta$MHinsol scenarios. In $\Delta$LNstate, precipitation $\delta^{18}O$ is the primary factor contributing to the differences in overall $\delta^{18}O_p$ and causes the negative anomalies across tropical South America, while differences in the annual cycle of precipitation amount show only a slight contribution (Fig. 5d–f). In $\Delta$MHinsol, precipitation $\delta^{18}O$ is the main cause of the negative anomalies in the northeast (Fig. 5h), whilst precipitation seasonality exerts a larger contribution to overall $\delta^{18}O_p$ over the western Amazon (Fig. 5i). To explain the spatial differences in $\delta^{18}O_p$, we examine how the annual cycle of precipitation and precipitation $\delta^{18}O$ is simulated over the western Amazon (70.3–78.8°W; 1.4–9.8°S) and eastern Amazon (59.0–47.8°W; 1.4°N–7.0°S) (Fig. 6). (Note that the eastern Amazon region is selected to include the Paraiso speleothem record (Wang et al., 2017).)

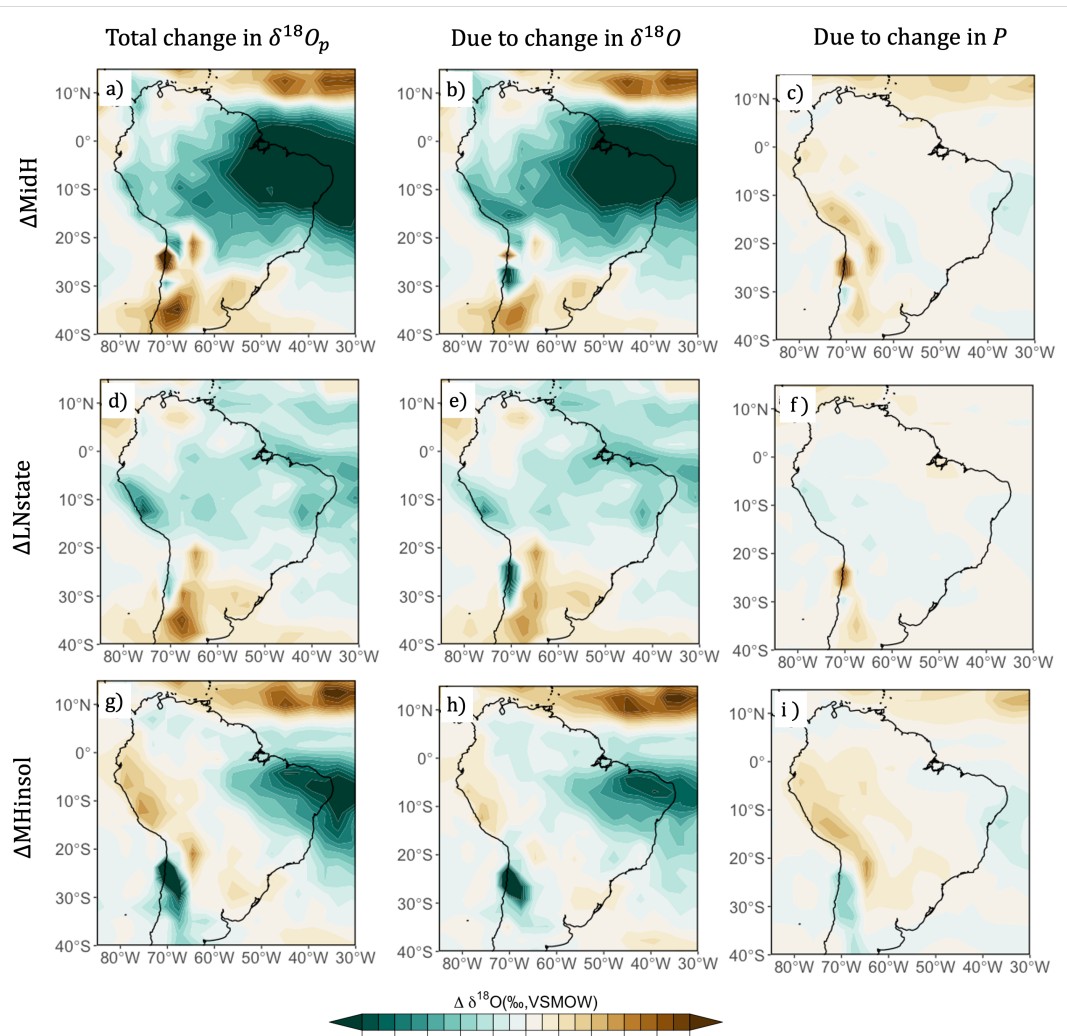

**Figure 5: Top row:** ΔMidH $\delta^{18}O$ for (a) the total $\delta^{18}O_p$ change [Eq. (1)], (b) $\delta^{18}O_p$ changes due to changes in the annual cycle of $\delta^{18}O$ of precipitation [Equation 2], and (c) $\delta^{18}O$ changes due to changes in the annual cycle of precipitation (P) [Eq. (3)]. **Middle and bottom row: Same as top row but for ΔLNstate and ΔMHinsol respectively.**

In the ΔLNstate scenario, the western Amazon sees significantly lowered precipitation rates around the austral autumn to winter (March to August). This reduction is driven by changes in the MAM zonal circulation (Fig. 4b), which strengthens subsidence over the western Amazon. Outside this period, however, precipitation rates show no significant change (Fig. 6a). The change in winter precipitation would result in overall lower $\delta^{18}O_p$ due to the reduced contribution of the relatively more positive $\delta^{18}O$ precipitation during winter. Moreover, precipitation $\delta^{18}O_p$ over the western Amazon becomes marginally more negative during both the winter and late austral summer months (Fig. 6c). This is likely due to an enhanced rainout effect over the upstream region in the east, which results in a larger influence on precipitation $\delta^{18}O$ (Fig. 5d–f). These combined effects lead to overall lower $\delta^{18}O_p$ values in the western Amazon in ΔLNstate compared to the control run.

In the northeastern region, the ΔLNstate scenario exhibits the strongest precipitation anomalies during the late austral summer to autumn (January to April). This coincides with an intensification of the ascending motion over the eastern portion of the continent (Fig. 4b), which leads to enhanced convective activity and higher precipitation rates in this region (Fig. 6b). Correspondingly, the stronger rainout effect results in more negative precipitation $\delta^{18}O$ during these months (Fig. 6d). This leads to an overall decrease in $\delta^{18}O_p$ values in the northeastern region as there is a larger contribution of more negative $\delta^{18}O$

summer precipitation to the overall $\delta^{18}O_p$ values. Consequently, both the east and western regions of the continent display negative $\delta^{18}O_p$ anomalies and a dipole pattern does not emerge in ΔLNstate.

Under the ΔMHinsol scenario, precipitation seasonality exerts a larger contribution to overall $\delta^{18}O_p$ over the western Amazon (Fig. 5i). Here, the annual precipitation seasonality is significantly reduced due to the weakening of the summertime Bolivian High system (Fig. 6c–d), resulting in an increased proportion of annual precipitation occurring during the winter months (JJA)

(Fig. 6a). The reduced contribution of the more $^{18}O$ depleted precipitation in DJF, as well as a larger contribution of the $^{18}O$ enriched JJA precipitation results in an overall $\delta^{18}O_p$ increase in the western Amazon. In the eastern Amazon, the annual cycle of precipitation remains relatively unchanged (Fig. 6b). However, the precipitation $\delta^{18}O$ is also lower during the late austral summer to autumn months, like the ΔLNstate scenario (Fig. 6d), thus, resulting in a similarly reduced $\delta^{18}O_p$ in the eastern region.

In summary, the experiments demonstrate that lower austral summer insolation and a more La Niña-like ENSO mean state can produce a dipole pattern in annual precipitation anomalies over tropical South America. However, the simulated $\delta^{18}O_p$ is influenced by both changes in precipitation $\delta^{18}O_p$ and shifts in seasonality. This distinction is particularly important in the western Amazon, where $\delta^{18}O_p$ reflects not only local precipitation amount but also the cumulative upstream rainout experienced by moisture transported inland from the tropical Atlantic. This complexity has also been emphasized by other studies that show

direct $\delta^{18}O$–precipitation interpretations can overlook the integrated isotopic processes that shaping continental signals (Orrison et al., 2022). In the northeast, both insolation changes and the ENSO mean state concurrently drive negative $\delta^{18}O$ values. In contrast, in the western Amazon, insolation-driven changes in precipitation seasonality result in a positive $\delta^{18}O$ response, whereas a more La Niña-like mean state alone leads to a negative $\delta^{18}O$ response, owing to the stronger influence of upstream-modified precipitation $\delta^{18}O$. Under the ΔMidH scenario, the combined effects of ΔMHinsol and ΔLNstate amplify

the negative $\delta^{18}O_p$ anomalies in the northeast. At the same time, their net impact likely counteracts the insolation-driven $\delta^{18}O_p$ signal in the western Amazon and Andes, resulting in a muted response in the west. Therefore, despite lower precipitation rates in the western Amazon in ΔMidH, $\delta^{18}O$ of precipitation itself still exhibits negative anomalies. Note that we cannot rule out the additional effect of the underestimated continental $\delta^{18}O$ gradient associated with the model's coarse resolution, which may slightly dampen the $\delta^{18}O$ response in the western Amazon. However, this bias arising from the coarse resolution would

primarily reduce gradient magnitude rather than invert the sign of anomalies, and thus is unlikely to explain the paradoxical $\delta^{18}O$–precipitation relationship identified here.

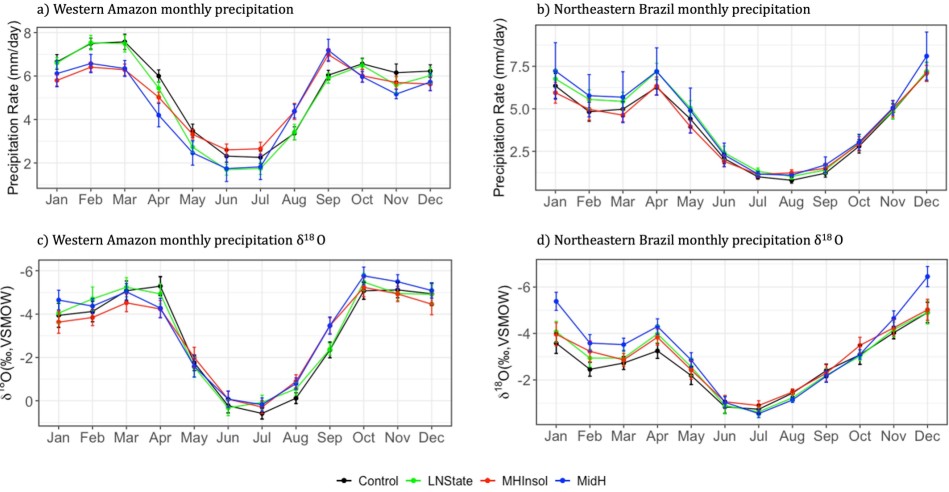

**Figure 6: Monthly precipitation rate (top row) and precipitation $\delta^{18}O$ (bottom row) for each experiment in the 'Control', 'LNstate', 'MHinsol', and 'MidH' experiment averaged over the western Amazon (70.3–78.8°W; 1.4–9.8°S) and eastern Amazon (59.0–47.8°W; 1.4°N–7.0°S).**

## 4.3 The role of global SST feedbacks in shaping the δ18O dipole

A key uncertainty remains regarding why the ΔMidH scenario does not fully capture the $\delta^{18}O$ pattern observed in proxy records. In this study, we prescribed mid-Holocene SST anomalies only in the tropical Pacific, based on the PMIP4 ensemble mean. This approach isolates the influence of Pacific SST changes but omits potential feedbacks from the global ocean. This omission is important because Atlantic SST variability is known to modulate the modern Amazonian precipitation through their effect on inland moisture transport (Ciemer et al., 2020; Yoon and Zeng, 2009). Studies of the SASM during the last millennium further shows that variations in the mean latitudinal position of the Atlantic ITCZ can exert a strong control on SASM strength, that operates alongside PWC changes (Lyu et al., 2024). These ITCZ-related changes influence not only the local precipitation amount at proxy-sites, but also the degree of upstream rainout over the tropical Atlantic (Lyu et al., 2024; Steinman et al., 2022), thereby influencing $\delta^{18}O$ signatures over the Amazon basin. Mid-Holocene Atlantic ITCZ also differed substantially from the present-day state. In particular, the mid-Holocene changes in Atlantic interhemispheric SST gradients likely produced a latitudinally broader ITCZ, which contracted towards the late-Holocene (Chiessi et al., 2021). Given the sensitivity of $\delta^{18}O$ to both changes in local precipitation and upstream rainout history, such mid-Holocene changes in the Atlantic ITCZ would be expectedly alter the isotopic composition of moisture into the continent. Therefore, the absence of Atlantic SST feedbacks, and specifically changes in ITCZ strength and width in the model simulations, likely contributes to the model-proxy mismatch observed in the MidH and LNstate scenarios.

This importance of Atlantic feedbacks becomes apparent when comparing our prescribed SST experiments to a mid-Holocene scenario using ECHAM4.6 coupled to a slab ocean model (Fig. S6a–b), which allows SSTs outside the tropical Pacific to respond dynamically. The slab ocean setup reproduces the observed $\delta^{18}O$ dipole across South America more successfully (Wong et al., 2023), suggesting that SST feedbacks may be important for capturing regional hydroclimate patterns. However, it underestimates the magnitude of $\delta^{18}O$ change in the Nordeste, likely due to the absence of full ENSO dynamics. A key distinction between the two simulations is the behaviour of the Atlantic ITCZ. The prescribed-SST configuration produces stronger MAM precipitation over the tropical Atlantic region (averaged between 60°W–0°; Fig S6c) relative to the slab-ocean setup. This likely reflects an overestimation of Atlantic ITCZ strength due to the absence of local ocean-atmosphere feedbacks which may act to dampen convection there. Consequently, an enhanced ITCZ convection in the prescribed-SST run would increase upstream rainout, resulting in more $^{18}O$-depleted moisture advected westward into tropical South America. This mechanism is consistent with more negative $\delta^{18}O$ in precipitation over the continent (Fig. 6b,d). In contrast, the slab-ocean simulation produces a weaker ITCZ response and $\delta^{18}O$ anomalies more consistent with proxy reconstructions. Together, the model comparisons suggest that the Atlantic ITCZ response may be essential for precipitation isotope changes in the western Amazon.

Another important difference is the experimental baseline: while the slab ocean experiments were run as mid-Holocene anomalies relative to a pre-industrial reference period (Wong et al., 2023), our ΔMidH experiments are computed relative to a present-day baseline, which would be warmer than the pre-industrial. This discrepancy in reference periods could partly explain differences in the estimated magnitude of mid-Holocene hydroclimate change between the two experiments. Additionally, we cannot rule out the possibility that the slab ocean may better capture certain aspects of mid-Holocene SST anomalies than the GCM-based PMIP4 ensemble mean used to prescribe Pacific conditions in the MidH experiment. However, since our experiments were not designed to disentangle the relative influence of spatial SST patterns from the role of global feedbacks, we cannot definitively attribute the improved performance of the slab ocean simulation to either specific factor.

## 4.4 ENSO impact on rectifying precipitation climatology over South America

The results from ΔNoENSO (Fig. 3d) shows the precipitation climatology in the absence of ENSO variability. This allows us to infer how the climatology of precipitation is rectified due to the presence of year-to-year variability in tropical Pacific SSTs despite the same long-term mean of SSTs.

Figure 3d illustrates that despite an equal change in the amplitude of variability in both El Niño and La Niña phases, ΔNoENSO shows only a marginal shift in precipitation patterns over tropical South America. Therefore, while a change in the ENSO amplitude of variance does not change the climatology of SSTs over the tropical Pacific, it may influence precipitation climatology elsewhere due to the non-linear impact of ENSO on precipitation.

We can extrapolate the ΔNoENSO response to infer how a muted ENSO variability during the mid-Holocene may have affected precipitation patterns. The reduction in ENSO variability produces a northeast–southwest dipole response that would

have contributed to the spatial pattern seen in hydroclimate records. However, the ΔNoENSO experiment represents an 'extreme' scenario of a 100% reduction of ENSO variability and thus sets an upper limit on how much the change in variability could affect precipitation patterns in contrast to the 20–60% reduction of ENSO variability that has been suggested during the mid-Holocene (e.g., Brown et al., 2020; Emile-Geay et al., 2016). As the 100% reduction in ENSO variability produced only a minor precipitation response relative to tropical Pacific mean state and insolation forcing, it is then unlikely that an even weaker reduction in the mid-Holocene would have been a dominant factor to account for the observed precipitation trends. We note that a similar observation regarding the impact of ENSO variability was previously found in the western Pacific region as well (Djamil et al., 2023).

## 5 Conclusion

In this study, we have examined the independent impacts of ENSO variability and ENSO mean state on the South American hydroclimate through a set of single-forcing experiments. We have firstly demonstrated that both a La Niña-like mean state and a lower SHSI are important drivers in shaping the east–west precipitation dipole response documented by regional proxy records during the mid-Holocene. In the western Amazon region, insolation changes have the greatest impact on precipitation rates during the peak monsoon months and the location of reduced subsidence in the insolation experiment is concentrated over the western tropical Atlantic (30°W). On the contrary, the La Niña-like mean state in the mid-Holocene results in the reduced subsidence over the Nordeste states to eastern Amazon (60°W). This difference in the location of reduced subsidence leads to distinct extents of precipitation change in the eastern region.

Based on our model-proxy comparison of $\delta^{18}O$, we found that during the mid-Holocene, relatively more positive $\delta^{18}O$ in precipitation over the western Amazon is predominantly governed by changes in precipitation seasonality due to the weaker SHSI. As a more La Niña-like mean state resulted in negative $\delta^{18}O$ anomalies — disagreeing with the effects of insolation and proxy records — we suggest that this effect is likely due to the lack of additional SST feedback from the global ocean. In particular, the prescribed-SST configuration in our simulations may have over-strengthened the Atlantic ITCZ in the absence of local ocean–atmosphere coupling, thereby enhancing upstream rainout which may have eventually led to a negative bias in precipitation $\delta^{18}O$ in the western Amazon. In contrast, in the northeast region, both seasonality and precipitation $\delta^{18}O$ (influenced by the amount effect) jointly control the overall $\delta^{18}O$ values. These findings highlight the complexity of interpreting $\delta^{18}O$ records in relation to past precipitation changes, emphasizing the importance of considering seasonality and amount effect in understanding the hydroclimate variability in the region during the mid-Holocene.

We furthermore demonstrate with the 'NoENSO' experiment, that the presence of modern ENSO variability has a rectifying effect on the climatology of precipitation over South America. However, the impact of a reduced ENSO variability is relatively minor in comparison to the effects resulting from changes in ENSO mean state and insolation. These findings imply that during the mid-Holocene, a reduced ENSO variability has a negligible contribution to precipitation trends over tropical to subtropical South America.

While our experiments provide first-order insight into how insolation forcing and ENSO mean-state changes affect $\delta^{18}O$ in precipitation, they also reflect the inherent limitations of an AGCM setup. In a fully coupled system, non-linear interactions between the ENSO amplitude and mean state—along with feedbacks from the Atlantic and other ocean basins—may further modulate the magnitude of the isotopic response. Nevertheless, the experimental design intentionally treats ENSO amplitude and mean state as separate boundary conditions, allowing us to isolate the sensitivities of precipitation and precipitation isotopes to each component, and thereby clarify their individual contributions to mid-Holocene hydroclimate change.

Current projections of future changes in ENSO variability and the east–west SST gradient over the tropical Pacific remain uncertain due to conflicting projections from different climate models (Cai et al., 2022; Kohyama et al., 2017; Poli et al., 2016). Given the considerable uncertainties surrounding the changes in these two ENSO characteristics in future climates, conducting individual forcing experiments such as done here is vital for understanding their distinct influences. Although this study primarily examines ENSO variability and mean state, this approach can be extended to assess the effects of other ENSO characteristics, for example, the degree of asymmetry between El Niño and La Niña phases, or a change in the periodicity of ENSO occurrence. Furthermore, these experiments may refine our interpretation of paleoclimate proxies, such as precipitation $\delta^{18}O$, and thus enable a better reconstruction of past hydroclimate change and the underlying mechanisms driving these changes.

**Data availability**

The results of the ECHAM4.6 model simulation in this study is available at (Wong, 2025). The ERA5 data set is available in Hersbach et al. (2020).

**Author contribution:**

MW and XW planned and designed the experiments; MW performed the experiments; MW analysed the data; MW wrote the manuscript draft; MW and XW reviewed and edited the manuscript.

**Competing interests**

The authors declare that they have no conflict of interest.

**Acknowledgements**

We are in debt to David S. Battisti for his generous contributions to the design and analysis of the experiments, as well as for providing valuable support with the computing resources. We also thank Xiaojuan Liu and Qinghua Ding for their helpful

advice and discussions at the various stages of this work. Finally, we are very grateful to the two anonymous reviewers for their constructive feedback and comments that significantly improved this manuscript.

**Financial Support**

This research was supported by the Earth Observatory of Singapore via its funding from the National Research Foundation Singapore (NRF), the Singapore Ministry of Education (MOE) under the Research Centres of Excellence initiative and MOE research grants (MOE-T2EP10122-0006  and MOE-MOET32022-0006 to X.W.). M.W. is supported by a Nanyang Graduate President's Scholarship and a Stephen Riady Geoscience Scholarship.

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
