# Peer review of "The relative impacts of tropical Pacific teleconnections and local insolation on mid-Holocene precipitation over tropical South America"

_EGUsphere, 2025_

## Author Comment (AC1)

**Reply to Referee #1**

We thank the Anonymous Referee #1 for the detailed review and very constructive comments. The comments of Referee #1 are provided in black text below and our replies to each comment are shown in the blue text.

This is an interesting paper discussing how changed ENSO characteristics may have affected South American climate during the mid-Holocene. Several model experiments are carried out to probe the sensitivity of the South American monsoon to Holocene ENSO characteristics and how these changes are imparted on the isotopic composition of precipitation across the region. The experiments are set up as sensitivity studies that allow diagnosing the influence of a changed ENSO mean state and changed insolation of ENSO on South American climate. The only drawback I can see with these experiments is that they are based on a rather old and outdated version of the ECHAM model, run at a very coarse resolution. The interpretation and description of the results will need some revisions and improvements as outlined below, but overall I think the paper is worth publishing after some moderate revisions.

Throughout the paper more care needs to be exercised with specific statements that are not clear or unambiguous. In the abstract, for example, there are several statements that need to be clarified as they are inconclusive when read on their own. For example, the authors write that' ENSO mean state changes suppress winter precipitation', yet it is unclear for which direction of mean state changes this is valid. Similarly, the statement that 'both SHSI and ENSO mean state changes directly influence precipitation  $\delta 18O$ , resulting in strong negative  $\delta 18O$  anomalies' is ambiguous as it is not clear whether positive or negative mean state changes lead to these negative d18O anomalies.

We appreciate the reviewer's comment and will make the changes accordingly to improve the clarity of these statements throughout the manuscript. In both statements mentioned here, it is specifically a La Niña-like mean state change that suppresses the SASM winter precipitation.

Figure 2: it might be better to refer to the more neutral term 'precipitation' and not 'rainfall' in all panels. Certainly for the description of the Huascaran ice core the descriptor 'rainfall' is inappropriate, as accumulation at that site is exclusively in the form of snow and not rain.

Agree. We will make the changes in the main text.

Figure 2: The location of the box termed 'northeastern Brazil' is outside the actual region known as northeastern Brazil. The box is located over the mouth of the Amazon river in northern Brazil, while northeastern Brazil is known as the region forming the 'knee of Brazil' that extends out toward the southern tropical Atlantic.

The 'northeastern Brazil' box was intentionally placed over the mouth of the Amazon River to encompass the location of the Paraíso Cave speleothem record for comparison purposes. To avoid confusion with the actual northeastern Brazil region, we will rename this box to 'eastern Amazon' in the revised figure and when referred to throughout the manuscript.

Figure 2: How exactly were wet, neutral and dry conditions determined? The Sajama ice core record for example is indicated as 'drier', yet the trend over the last 6 ky in that record is flat. In fact Table S2 confirms that the values for MH and present-day are essentially identical. This comment also applies to Figure 3a-d.

We thank the reviewer for noting this ambiguity that requires clarification. The interpretation of mid-Holocene conditions in Table S2 (specifically column 6), and the proxy colours in Figure 3a-d are based on each author's conclusions in each paper, so that there is no further interpretation imparted on the climate records (this was done following Prado et al., 2013, to extend the records beyond the northeastern Brazil records in that paper). In the case of the Sajama ice core, Reese et al. (2013) interpreted low pollen abundance in the ice core as dry conditions during the mid-Holocene (specified as 8.0 to 5.5 Ka BP and thus consistent with the definition of 7.0-5.0 Ka BP in our manuscript), and the pollen concentration rising after that period was interpreted as wetter conditions to the present. Hence, although there is only a small magnitude of d18O change in the Sajama record, the precipitation direction is interpreted as 'dry' during the mid-Holocene. This clarification on how the direction of change was determined will be added to the revised Supplementary Text. However, we do not believe that this would significantly change the interpretations of the paper as most of the Andean records (not only speleothems but also the Huascaran ice core and Laguna Pumacocha sediment record) located along the western Amazon are interpreted with drier conditions which agrees with the positive d18O difference in the mid-Holocene.

The model version used, ECHAM 4.6, is rather old and outdated. The spatial resolution is very low. I don't know if results with a newer version would be fundamentally different, but it is a bit of a concern. I think the failure of the model to accurately reproduce the mid-Holocene conditions over the western Amazon and tropical Andes (Fig. 3e) may partly be attributed to the low resolution and lack of resolving topography over the region. There is not much the authors can do about this aspect, but it should be mentioned somewhere in the paper.

We understand the reviewer's concern regarding the model version and resolution. We agree that the ECHAM 4.6 is relatively old, however, this version was used because it is isotope-enabled, which is essential for our study that directly compares model simulation with isotope proxy records. We also agree that the coarse resolution likely limits its ability to accurately represent the topography towards the Andes. This limitation has been acknowledged in Section 3.1 (Validation, lines 195-200), and we have also compared the performance of ECHAM 4.6 with a higher resolution simulation (Fig. S4), which demonstrates that the model does underestimate the d18O decrease over the western Amazon. However, in our analysis we focus on the relative differences (anomalies) between each experiment and the baseline, which should minimize the direct impact of this systematic bias across scenarios (although we recognize the relationship is not strictly linear). We will expand on this discussion in the validation section to clearly state this caveat.

Why is ERA5 plotted in such low resolution in all Figures? Is it upscaled for better comparison with the model results?

ERA5 was interpolated to the model resolution for consistency with the model results.

Discussion Figure 3: As the authors correctly state on lines 221-222, 'The difference between the 'MidH' and 'Control' simulation, referred to as 'ΔMidH' should closely approximate the mid-Holocene conditions as captured by regional proxy records'. Yet the model clearly fails to reproduce the observed mid-Holocene enrichment seen in the proxies over the western Amazon and the Andes (Fig. 3e). Even though the signal is better reproduced in the MHinsol signal, this still points to a model deficiency in reproducing Mid-Holocene conditions over the tropical S. America region. This needs to be acknowledged somewhere in the text. The same comment applies to the La Nina state simulation which appears to produce dry conditions over the

western Amazon and tropical Andes (Fig. 3b), even though this region experiences excess precipitation during La Nina events. Hence I am a little bit worried about using these simulations to draw conclusive inferences about the relative roles of La Nina state vs. insolation in affecting precipitation and d18O in the region. I don't think this completely invalidates the results, but a more cautionary tone in the discussion and conclusions, better acknowledging the model deficiencies and caveats seems appropriate.

We agree that the  $\Delta$ MidH experiment does not fully reproduce the d18O enrichment observed in the western Amazon and Andean proxy records and will state it more explicitly in the text. In the revised text we will emphasize this mismatch in Section 3.4 and discuss it in further detail in Section 4.2 and 4.3.

Regarding the La Niña-state experiment, we acknowledge that the simulated drying over the western Amazon contrasts with some tendency toward wetter conditions observed during present-day La Niña events. As the reviewer notes, the coarse resolution of the model likely contributes to an underestimation of d18O decrease in this region, a bias we have acknowledge in the manuscript. However, we consider it unlikely that this systematic bias alone could invert the sign of the d18O anomaly. Instead, we suggest that the paradoxical signal may stem from the experimental set up on the LNstate scenario, in which only tropical Pacific SST anomalies were prescribed without global SST feedbacks (which we will elaborate on in the Discussion, Section 4.3).

Furthermore, there is a less prominent relationship between La Niña events and western Amazon hydroclimate — This is suggested by the ERA5 precipitation anomaly correlations with Niño 3.4 shown in Fig. S5b, which is not as significant in the western Amazon compared to other areas like northeastern Brazil and southeastern South America. This is also observed in analysis of precipitation anomalies over South America by Cai et al. (2020). Thus, while the coarse resolution likely amplifies the anomaly, the unexpected direction is more plausibly linked to the other reasons we have suggested. Overall, we do agree that this is an important point for readers to note when understanding the interpretations of the work, and will rephrase it to adopt a more cautious tone and also highlight this caveat earlier in the work.

Also, it is not clear to me why this analysis is carried out on an annual basis. Almost all proxy sites analyzed are located in the monsoon region and heavily biased toward the austral summer season. ENSO is similarly phase-locked seasonally. So why not focus on this season? It would mostly likely provide for a much cleaner diagnosis.

Our focus on annual precipitation-weighted d18O is motivated by the intention to directly compare the model output with proxy records, which generally integrate annual precipitation, particularly for speleothems which make up the bulk of the proxy records used. Following the reviewer's suggestion, we plot here the calculated precipitation-weighted d18O anomalies for the austral summer (DJF) season alone. These results (Figure 1 here) show broadly similar spatial patterns to the annual precipitation-weighted mean anomalies, albeit with a larger amplitude of change. Thus, we stick to using the precipitation-weighted mean (Eq. 1) as this accounts for the seasonal distribution of rainfall and thus already incorporates the influence of the summer season weightage.

Figure 1. Precipitation-weighted d18O over the austral summer season (December to February) for the(a)  $\Delta$ MidH, (b)  $\Delta$ LNstate, and (c)  $\Delta$ Insol.

Figure 6: same comment as above. Note that what is plotted here for 'northeastern Brazil' is really precipitation over the mouth of the Amazon. Northeastern Brazil should be characterized by a clear MAM precipitation peak.

Noted on the misnomer. We will revise all references to 'northeastern Brazil' box as 'eastern Amazon' instead of 'northeastern Brazil'.

Discussion section 4.3. the argument that SST feedbacks outside the Pacific also matter for South American climate and d18O signals is well taken. In fact, this was shown in recent analyses by Steinman et al. (2022) and Lyu et al. (2024), both documenting the joint influence of Pacific and Atlantic in modulating past d18O signals over tropical South America. A more thorough discussion of this aspect seems warranted here, as currently this section is rather speculative and not fully incorporating the latest scientific findings on this aspect.

We thank the reviewer for this valuable comment and agree that our discussion of the Atlantic influence on South American hydroclimate and d18O variability can be expanded here and better grounded in recent literature. In the revised manuscript, we will include a more detailed discussion particularly on the role of Atlantic SSTs, drawing on the studies suggested by the reviewer that emphasize the joint Pacific and Atlantic control on the SASM.

Our revision will also add on how our own model results are consistent with this view. Comparing the prescribed-SST (MidH experiment in this study) and a mid-Holocene scenario using the same model but with a coupled slab-ocean (as shown in Figure S6), we find that the prescribed-SST MidH setup simulates stronger MAM precipitation over the tropical Atlantic ITCZ region (60°W–0°) (Figure 2 here), than the run using a coupled slab-ocean model, which produces a comparatively weaker ITCZ. This suggests that the prescribed-SST configuration may be overestimating Atlantic ITCZ strength due to the absence of local ocean feedbacks which may act to dampen convection. Consequently, enhanced ITCZ convection in the prescribed-SST run likely leads to greater upstream rainout and more lighter d18O in the moisture advected westward into tropical South America, contributing to the lighter d18O anomalies seen over the western Amazon seen in the MidH experiment. This is in contrast to the coupled slab-ocean simulation which produces a weaker Atlantic ITCZ response and d18O anomalies that are more consistent with proxy reconstructions.

These findings corroborate that Atlantic feedbacks play a key role in modulating the isotopic response to Pacific forcing, and is also consistent with evidence that mid-Holocene Atlantic

ITCZ variability was sensitive to interhemispheric SST gradients (e.g., Chiessi et al., 2021). We will revise Section 4.3 to include this discussion and to reflect more explicitly how both Pacific and Atlantic feedbacks jointly influence the hydroclimate and isotopic patterns simulated for the mid-Holocene.

Figure 2. Averaged (between 60°W and 0°) precipitation for March–April (in red) and August–September (in blue) for the mid-Holocene prescribed SST experiment (MidH) in solid line, and for the mid-Holocene scenario using the same model coupled to a slab-ocean model in dashed line (Slab-ocean MH).

In the supplement (Section S2.), it is stated that no changes are expected in teleconnections from the Pacific to South America affecting d18O in precipitation over South America during the historical period. While there may be no significant trends, it is well known that Pacific multidecadal variability significantly modulates the d18O signal over South America (e.g. see recent analysis by Orrison et al. 2024). So the choice of the time period used as baseline for this analysis does matter. Furthermore, most d18O records over tropical S. America show a clear increase in the d18O values after 1850 CE (in many papers this is referred to as the Current Warm Period, or CWP, e.g. see Bird et al. 2011), hence in many records the values over this period are significantly more enriched in 18O compared to the prior preindustrial period from 850-1850 CE.

We agree that our original statement in the Supplement was too definitive, as multidecadal modes of variability in both the Pacific and Atlantic basins can indeed modulate d18O in precipitation (Orrison et al., 2024), even in the absence of strong long-term precipitation trends over tropical South America.

Regarding the d18O signatures during the CWP, we acknowledge that some records, such as the Laguna Pumacocha record mentioned by the reviewer, show an increase in d18O during the CWP. This enrichment would influence the absolute d18O values when compared to the pre-industrial period, as shown in Table S2. However, our primary focus is on the relative

differences between the mid-Holocene and the pre-industrial period, rather than between the CWP and the pre-industrial. The magnitude of change between the mid-Holocene and the pre-industrial still remains substantially larger than that between the pre-industrial and CWP. As shown in Table S2, the magnitude and direction of d18O change are consistent regardless of whether the pre-industrial or historical periods are used as the reference. Only one record (with only a single sample point within the Historical period) shows a reversal in the direction of change, and several records have limited sample coverage during the historical interval. For this reason, using the pre-industrial period as the reference provides a larger and more robust dataset for comparison, while preserving the overall directionality of d18O change across sites.

We will revise this section accordingly to better reflect this nuance, clarifying that while short-term variability may modulate d18O, the relative differences between the mid-Holocene and the pre-industrial remain a robust indicator of long-term hydroclimate change.

Figure S5 panel d). In the heading is states that this panel shows the correlation between GNIP d18O and the Nino3.4 index. Yet in the caption it is stated that the panel shows correlations with proxy data. Which is it and how would a correlation based on proxy data be calculated?

The sentence refers to the fact that the GNIP stations used and plotted in Figure S5d only covers the region of focus for this study, which is the tropical South America region where our proxy data covers in this study. To clarify, this shows the correlations of precipitation d18O from GNIP stations with Nino 3.4 index, not proxy record d18O. We will rephrase this in the text for clarity.

Figure S5: What are the gray cells in Figures S5e & S5f showing? I assume they indicate percentages above 200% difference (since they are in the middle of the red ITCZ region and are apparently showing significant changes (the cells are stippled). They should be plotted using saturated red colors, not an unexplained gray color.

The gray cells are values over 200% difference, we will replot this with saturated colours.

Minor edits:

Line 76: a more 'La Nina-like' state of the tropical Pacific

Line 241: 'Numbers next to speleothem sites'. You also show ice core and lake sediment records in this Figure, so you should refer to 'Numbers next to proxy sites' here.

Line 465: 'Lawrence' is a first name and should be abbreviated

Line 505: no need to capitalize 'J. Atmos. Sci.'

Line 512: check formatting of the tilde sign (El Nino)

Supplement Line 16: Pacific

Supplement Line 91: is run

Supplement Line 102: '(d) ERA5 Reanalysis data' should be '(f) ERA5 Reanalysis data'

Supplement Line 111: 'Francisco' is a first name and should be abbreviated

We appreciate the reviewer for pointing out these errors and shall correct them in the revised manuscript.

**References cited in review**

Bird, B.W., et al., 2011: A 2,300-year-long annually resolved record of the South American summer monsoon from the Peruvian Andes. *Proc. Nat. Acad. Sci.*, 108(21), 8583-8588, https://doi.org/10.1073/pnas.1003719108.

Lyu, Z., et al., 2024: South American monsoon intensification during the last millennium driven by joint Pacific and Atlantic forcing. *Sci. Adv.* 10, eado9543, https://doi.org/10.1126/sciadv.ado9543.

Orrison, R., et al., 2024: Pacific interannual and multidecadal variability recorded in δ18O of South American Summer monsoon precipitation *J. Geophys. Res.*, 129(17), e2024JD040999, https://doi.org/10.1029/2024JD040999.

Steinman, B.A., et al., 2022: North-south antiphasing of neotropical precipitation over the past millennium. Proc. Natl. Acad. Sci., 119(17), e2120015119, https://doi.org/10.1073/pnas.2120015119.

**Additional references cited in the response**

Prado, L.F., et al., 2013. A mid-Holocene climate reconstruction for eastern South America. *Climate of the Past*, 9(5), 2117-2133, https://doi.org/10.5194/cp-9-2117-2013.

Reese, C.A., et al., 2013. An ice-core pollen record showing vegetation response to Late-glacial and Holocene climate changes at Nevado Sajama, Bolivia. *Annals of Glaciology*, 54, 183-190, https://doi.org/10.3189/2013AoG63A375.

Cai, W., et al., 2020. Climate impacts of the El Niño–Southern Oscillation on South America. *Nature Reviews Earth & Environment*, 1, 215- 231, https://doi.org/10.1038/s43017-020-0040-3.

Chiessi, C.M., et al., 2021. Mid- to Late Holocene Contraction of the Intertropical Convergence Zone Over Northeastern South America. *Paleoceanography and Paleoclimatology*, 36, e2020PA003936, https://doi.org/10.1029/2020PA003936.

---

## Author Comment (AC2)

**Reply to Referee #2**

We thank the Anonymous Referee #2 for the thoughtful review and very constructive comments. The comments of Referee #2 are provided in black text below and our replies to each comment are shown in the blue text.

The manuscript explores how changes in the ENSO characteristics and local solar insolation during the mid-Holocene influenced precipitation patterns and oxygen isotope ratios in tropical South America, using a water isotope-enabled atmospheric general circulation model (ECHAM4.6).

The authors use climate simulations to understand the effects of: (1) Reduced ENSO variability, (2) A La Niña-like mean state in the tropical Pacific, (3) Lower Southern Hemisphere summertime insolation (solar input due to orbital changes).

The major finding includes: (1) Reduced ENSO variability had only minor effects on average precipitation and isotope ratios. (2) A La Niña-like mean state and lower insolation both produced an east-west dipole in rainfall changes (drier western Amazon/southern Brazil, wetter northeastern Amazon/Nordeste region), consistent with a number of paleoclimate records. (3) However, changes in isotope ratios did not always mirror precipitation changes due to differences in how precipitation seasonality and mean state changes affected regional isotope records. (4) The model indicates that western Amazon isotope anomalies during the mid-Holocene are more strongly influenced by weaker insolation (seasonality) than by Pacific mean state changes, whereas both factors reinforce strong negative isotope anomalies in the northeast.

I actually really like this "clean" approach but having some concerns about the experiment design. I hope that the following points can be addressed for better understanding and clarity.

Model configuration and experiment design: The study uses the ECHAM4.6 atmospheric general circulation model with prescribed SSTs and runs several experiments manipulating ENSO characteristics and insolation. The decision not to use a fully coupled ocean-atmosphere model in the main analysis may limit the realism of certain feedbacks, especially given the demonstrated importance of global SST feedbacks in shaping isotope patterns (kind of circular). The rationale and limitations of prescribing only tropical Pacific SST anomalies, versus allowing full ocean dynamical feedback, should be explicitly justified.

We thank the reviewer for highlighting this important point regarding the justification for using an AGCM. We used an AGCM with prescribed SSTs because this configuration provides a controlled and computationally efficient way to isolate and cleanly quantify the atmospheric response to specific changes in the tropical Pacific SSTs. We believe this approach is suited for single-forcing sensitivity experiments, where the objective is to examine the direct atmospheric response to a specific boundary condition. Nevertheless, we acknowledge that this approach is closer to an idealised scenario as the lack of a dynamic ocean-atmosphere feedback omits the full range of feedbacks which may influence the long-term variability, and is likely one of the reasons for the mismatch between the LNstate scenario with proxy records. To address this, we will add a clear justification of using an AGCM in the introduction and the implications of how it may differ from a full GCM.

Proxy-Model comparison (Western Amazon): The model fails to replicate positive isotopic anomalies recorded in Western Amazon proxies. More discussion or attempted quantification of the likely causes (missing Atlantic SST feedbacks, proxy uncertainties, limitations of isotope parameterizations or even dynamical reasons) would be valuable for both paleoclimate and modeling audiences.

We agree that the manuscript requires elaboration on the causes for the model-proxy discrepancy in the western Amazon. In the manuscript, we note that the discrepancy is likely attributed to the absence of SST feedbacks, particularly over the Atlantic. We will expand this discussion section (Section 4.3) in the revised manuscript to incorporate more recent literature and to explore these mechanisms in more detail. Specifically, we will show how a comparison between our prescribed-SST experiment and a complementary run using the same model coupled to a slab ocean indicates that the prescribed-SST configuration produces a stronger mid-Holocene Atlantic ITCZ and enhanced precipitation over the tropical Atlantic. The difference likely arises from the missing local ocean feedbacks that would otherwise dampen the convection there. The resulting stronger ITCZ in the prescribed-SST run thus increases the upstream rainout effect, resulting in more depleted d18O in the moisture advected into the western Amazon. This mechanism explains the negative d18O anomalies in the western Amazon despite lower simulated precipitation rates there.

In contrast, a coupled-ocean experiment produces a weaker mid-Holocene ITCZ response and d18O anomalies that are more consistent with proxy records. Although the slab-ocean configuration still lacks the full dynamical feedback of a fully coupled ocean model, even the partial inclusion of surface ocean coupling nevertheless substantially reduces the model-proxy mismatch, highlighting the importance of ocean feedbacks.

We also note that while factors including the proxy uncertainties and model biases could influence the magnitude of the simulated-proxy differences, they are unlikely to fully account for the directional mismatch in the western Amazon. For instance, systematic model biases are expected to remain similar across experiments, and therefore they would influence the amplitude of d18O rather than the directional sign of the isotopic response. Moreover, they do not account for counterintuitive pattern in the western Amazon of lower precipitation accompanied by more negative d18O anomalies, which is more plausibly explained by the Atlantic ITCZ mechanism above (i.e., a change in the d18O of the upstream moisture source).

Separation of ENSO magnitude and mean state: The experimental design cleanly separates mean state and ENSO amplitude effects, but real-world ENSO regimes often exhibit covariation and nonlinearities. Please elaborate on how robust these attributions are, and discuss any residual uncertainty in interpreting main findings as the result of independent factors.

Our experimental design treats ENSO amplitude and mean state as separate boundary conditions to isolate the first-order atmospheric and isotopic responses to each component. We acknowledge, however, that in the real climate system these components are coupled and can covary. For example, changes in ENSO variability over the recent decades have been linked to shifts in the background state of the tropical Pacific (e.g., Chung and Li, 2013; Lübbecke et al., 2014). Therefore, in a fully coupled system, non-linear interactions between the ENSO amplitude and mean state may modify the full magnitude of the isotopic responses simulated here.

Therefore, our experiments can be view as sensitivity test that examine the direction and relative strengths of atmospheric and isotopic responses, rather than equilibrium outcomes of a fully coupled climate system. We will include this caveat in the introduction section to better contextualise the scope and interpretive limits of our findings, and emphasize that our results illustrate the mechanistic pathways linking each forcing to isotopic responses, while recognizing that the absolute magnitude and spatial patterns of these responses may differ in a fully coupled system.

Isotope dynamics and proxy interpretation: The manuscript acknowledges that the  $\delta^{18}$ O rainfall signal is complicated by competing effects of precipitation amount, seasonality, and source moisture. This section would benefit from a more dynamical explanation to show how the induced circulation plays a role. Also The ECHAM4.6 runs at a relatively coarse resolution, which may dampen key gradients and convective responses. The implications for hydroclimate signal fidelity should be discussed more explicitly.

We agree that a more explicit discussion of how modified circulation plays a role in shaping the d18O signal will strengthen this section. We will provide more detail in Section 4.2 to relate the interpretation of the d18O anomalies more directly to the underlying changes in the circulation dynamics, in order to extend the discussion beyond the local precipitation amount effect.

Specifically, we will better detail how the changes in both precipitation and precipitation d18O in the eastern and western Amazon (shown in Figure 6) are tied to changes in the zonal circulations (such as the changes in Walker Circulation discussed in section 4.1) and how this modulates d18O thorough both the amount effect as well as the source moisture. In the LNstate experiment, for instance, the enhanced ascending motion over during DJF-MAM over the eastern Amazon, associated with a shifted Walker Circulation, intensifies convective rainout, producing more depleted d18O in precipitation upstream. The concurrent simulated strengthening of the Atlantic ITCZ further increases upstream rainout over the tropical Atlantic, which depletes the d18O of the moisture advected westward into the Amazon Basin. Conversely, in the ΔMHinsol experiment, weaker continental convection (during DJF) reduces local rainout in the northeast and enhance the contribution of isotopically heavier Atlantic-sourced moisture, resulting in comparatively enriched d18O anomalies.

Regarding the model resolution, the coarser resolution used in the model leads to a reduced continental gradient of d18O towards the west, likely due to the underestimation of topographic influence near the Andes. This limitation has been evaluated and explicitly noted in the validation section. While the coarse resolution may dampen the local hydroclimate and isotopic variability, we emphasize that our analysis focuses on relative anomalies between experiments, which minimizes the influence of systematic biases. We will make these caveats and implications clearer in the revised discussion.

**References cited in the response**

Chung, P., and T. Li, 2013: Interdecadal Relationship between the Mean State and El Niño Types. J. Climate, 26, 361–379, https://doi.org/10.1175/JCLI-D-12-00106.1.

Lübbecke, J. F., and M. J. McPhaden, 2014: Assessing the Twenty-First-Century Shift in ENSO Variability in Terms of the Bjerknes Stability Index. J. Climate, 27, 2577–2587, https://doi.org/10.1175/JCLI-D-13-00438.1.